# Surveying Parents’ Awareness and Adherence to Influenza Vaccination Recommendations in Children in Southern Italy

**DOI:** 10.3390/vaccines10081298

**Published:** 2022-08-11

**Authors:** Gabriella Di Giuseppe, Giovanna Paduano, Sara Vaienna, Giuseppe Maisto, Concetta Paola Pelullo, Maria Pavia

**Affiliations:** 1Department of Experimental Medicine, University of Campania “Luigi Vanvitelli”, Via Luciano Armanni 5, 80138 Naples, Italy; 2Department of Movement Sciences and Wellbeing, University of Naples “Parthenope”, Via Medina 40, 80133 Naples, Italy

**Keywords:** children, influenza vaccination, Italy, survey, vaccination adherence

## Abstract

Background: This study explored awareness and adhesion of parents to the recommendations for influenza vaccination for their children. Methods: This cross-sectional study was conducted from March to June 2022 among parents of 6 months–6 years-old children in Southern Italy. Results: Only 22.1% of parents were aware that influenza vaccination was recommended for healthy children, and 31.8% reported at least one influenza vaccination in the four influenza seasons from 2018/2019 to 2021/2022. Adherence was higher among parents who had received influenza vaccination at least once, who knew the main childhood vaccinations, who knew that influenza can be prevented, that influenza vaccination is recommended for healthy children, who believe that it is useful, and who have been informed about it by physicians. Willingness to vaccinate their children in the upcoming season was 48.6% and was higher among parents who had vaccinated their children at least once, believed that influenza vaccination protects against severe complications, believed that it is useful, and that it is not better for children to acquire immunization through illness than through vaccination. Conclusions: These findings suggest that efforts should be devoted to educating parents on the risks associated with the disease and the benefits of the vaccine.

## 1. Introduction

Influenza is a globally spread disease and the leading cause of seasonal epidemics, worldwide. Influenza may affect people of any age, but complications and hospitalizations are more frequent in childhood, especially in children aged ≤ 5 years, people with underlying chronic clinical conditions, and people aged ≥ 65 years [1,2,3]. Recent estimates on the burden of influenza have shown that worldwide, in children less than 5 years of age, in 2018 there were 109.5 million cases of influenza, almost 10 million cases of influenza-associated acute lower respiratory infections (ALRI), and about 870,000 hospital admissions, 15,000 in-hospital deaths, and around 35,000 total deaths from influenza virus-associated ALRI [4].

Moreover, there is consistent evidence that children play a leading role in propagating influenza A epidemics to other age-groups of the population [5,6], including older people, who are at higher risk of severe influenza complications [7]. Several studies have demonstrated, with different methods, that children are the first population group to be affected by influenza at the onset of the epidemic, and are the “drivers” to transmission to other groups [6,8], and that their vaccination would reduce significantly the spread in the community [5,9]. Vaccination is the most effective method to prevent and control the spread and clinical complications of influenza [7,10,11,12], and all countries have developed seasonal vaccination strategies to contrast the spread of influenza, particularly in most vulnerable subjects. In Italy, every year the Ministry of Health produces recommendations, indicating which groups of the population should receive influenza vaccination. In order to achieve the specific objectives of the influenza immunization program, influenza vaccination is actively offered free of charge to those included in the Ministry of Health recommendations. Furthermore, after having vaccinated the eligible population groups, if there are excess stocks of the vaccine, it is offered free of charge to anyone who requests it. Beginning with the 2020/2021 season, vaccination in healthy children aged 6 months–6 years was recommended, whereas before that season the recommendation included only children with underlying chronic medical conditions [13]. The rationale for this new recommendation was to have the opportunity to reduce the spread of influenza in children during the COVID-19 pandemic and, thereby, in adults and the elderly [14,15,16]. Therefore, many Italian regions have begun to offer this vaccination to healthy children, in collaboration with general practitioners and pediatricians. 

Several influenza vaccines are available for children, including the subunit quadrivalent split (QIV) vaccine for children from 6 months of age, the quadrivalent vaccine on cell cultures (VIQcc) for children from 2 years of age, and, from the 2021/2022 influenza season, a nasal spray of live attenuated vaccine (LAIV) for children from 2 years of age [15,16]. 

Although recommended, influenza vaccination continues to be under-used, particularly in subjects with underlying chronic clinical conditions, and, to our knowledge, there are scarce data on adherence to the new strategy involving healthy children in Italy [17]. Indeed, the most recent data on influenza vaccination coverage have shown, in the period before the COVID-19 pandemic, that among children the highest adherence was reported in 2 to 4 years-old subjects both in Italy (4.2%) and in Campania region (4.7%) [18]. Moreover, a substantial increase in influenza vaccination coverage was detected during the COVID-19 pandemic in all Italian regions with the highest value in 2 to 4 years-old children (17.4%), whereas in Campania it was in 2 to 4 and 5 to 8 years-old children (17%) [19].

Since it has been reported that during the COVID-19 pandemic there has been, among parents, an increase in interest and a shift towards more favorable attitudes regarding vaccinations [17,20,21], this study was undertaken to explore the level of awareness and adhesion of parents to the recommendation of influenza vaccination in their healthy children aged 6 months to 6 years. This new knowledge would be of interest for the development of interventions to improve participation of this group of the population to a strategic vaccination campaign. As a secondary objective, other vaccinations recommended for children were investigated.

## 2. Materials and Methods

### 2.1. Study Population, Sampling Procedure and Data Collection

This survey study was undertaken from March 15 to June 30, 2022, and involved parents of 6 months to 6 years-old children attending nurseries and kindergartens in an area of Southern Italy (Campania region). 

A two-stage sampling design was conceived. Nurseries and kindergartens were randomly selected from a list of all the 120 public and private facilities in the Naples metropolitan area, and all eligible parents of children attending the selected structures were invited to participate in the study. To be eligible, children had to be at least 6 months old and not older than 6 years at the beginning of the 2021/2022 influenza season. The sample size was calculated by using a single population proportion formula, with the assumption that 50% [22,23] of parents had vaccinated their children against influenza, using a confidence level of 95%, a margin of error at 5%, and considering a response rate of 60%. Therefore, at least 640 parents should be invited for a final sample size of 384 parents. Details of the calculation are reported in the Appendix A.

Before starting data collection, the project was presented to the director of the involved facilities to obtain permission and collaboration. Once adhesion to the project was obtained, parents/guardians received a letter, an informed consent form and a questionnaire for self-administration. The letter briefly described the study aims and its relevance, inviting only one parent to complete the questionnaire. Participants were assured that adhesion to the project was voluntary, that confidentiality of responses would be granted, and were invited to fill and return the questionnaire and the signed consent form. To maintain confidentiality, the questionnaire and the consent form were received separately, so that no information concerning the identity of the respondent could be retrieved from the completed questionnaire. To improve the response rate, several follow-up reminders were provided by the directors to the selected parents.

The study protocol was approved by the Institutional Ethical Committee (0022399/i 23 July 2021).

### 2.2. Survey Instrument

A self-administered structured questionnaire was outlined and developed adopting the experience of previously published studies [20,22,24]. 

The questions were grouped into four sections: (1) socio-demographic and anamnestic characteristics of parents and children; (2) knowledge of recommended childhood vaccinations, including influenza, and of modes of transmission and prevention of influenza, as well as of population groups who are recommended to receive influenza vaccination; (3) attitudes and behaviours related to childhood vaccinations overall, and specifically to influenza, and (4) sources of information on influenza vaccination and need of additional information. The questionnaire is available in the Appendix A. To determine the level of knowledge about the main mandatory and recommended vaccinations in 6 months to 6 years-old children, an overall knowledge score was constructed considering 1 point for each correct answer, with a total knowledge score ranging from 0 to 10. Then, the median knowledge score was calculated and parents with a score above the median were considered to have accurate knowledge. Beliefs about vaccinations were measured on a 3-point Likert-type scale with options for agree, uncertain, and disagree. Concern about the risk for their children of contracting influenza and to have severe complications of the disease were measured on a 10-point Likert scale ranging from 1 (not at all worried) to 10 (extremely worried), and beliefs about the usefulness of the influenza vaccine for children on a 10-point Likert scale ranging from 1 (not at all useful) to 10 (extremely useful). For purposes of analysis, concern that their children might contract influenza and about influenza vaccine side effects, and belief that influenza vaccine is useful, were dichotomized as 1–7 (not worried/useful) = 0 and 8–10 (very worried/useful) = 1. Childhood vaccination uptake was measured by marking a list of several vaccinations, and for influenza by marking the list of the four influenza seasons from 2018/2019 to 2021/2022. Finally, the intention to vaccinate their children in the upcoming 2022/2023 influenza season was proposed with “yes” or “no” options and a list of potential reasons for having or not this intention was provided, allowing the marking of one or more reasons.

A pilot study among 50 parents was carried out to evaluate the readability, clarity, and correct flow of the items. As a result of the pilot study, the order of two questions was modified to improve the flow of reading, and the questionnaires involved in the pilot study were not included in the final survey. 

### 2.3. Statistical Analysis

All analyses were performed with Stata software version 15 [25]. First, descriptive statistics were conducted to summarize the main characteristics of the sample. Second, a univariate analysis was performed using the chi-square test and Student’s t-test to evaluate the association between several potential determinants and (1) having vaccinated at least once their children in the four influenza seasons from 2018/2019 to 2021/2022 (no = 0; yes = 1), and (2) willingness to vaccinate their children in the upcoming 2022/2023 influenza season (no = 0; yes = 1). In the multivariate logistic models the following independent variables, which were judged to potentially have influence on the above mentioned outcomes (Models 1 and 2), were included: gender of the parent (female = 1; male = 2); age of the parent, in years (≤35 = 0; 36–40 = 1; 41–45 = 2; >45 = 3); marital status (single parent = 0; married/cohabitant = 1); occupation (unemployed = 0; employed = 1); education level (primary, middle and high school = 0; university degree/master = 1); having received influenza vaccine at least once in the four influenza seasons from 2018/2019 to 2021/2022 (no = 0; yes = 1); number of children (ordinal) (1 = 1; 2 = 2; ≥3 = 3); children’s age, in years (≤3 = 0; >3 = 1); children’s chronic disease(s) (no = 0; yes = 1); at least one visit by the primary care pediatrician/physician in the previous 12 months (no = 0; yes = 1); knowledge of influenza modes of transmission (other = 0; droplet = 1); knowledge that influenza can be prevented by vaccination (no = 0; yes = 1); knowledge that influenza vaccination is recommended for healthy 6 months to 6 years-old children (no = 0; yes = 1); knowledge about the main mandatory and recommended vaccinations for 6 months to 6 years-old children (no = 0; yes = 1); belief that it is better for children to acquire immunization through illness than through vaccination (agree/uncertain = 0; disagree = 1); belief that there are too many vaccinations for children (agree/uncertain = 0; disagree = 1); confidence in the information about vaccinations provided by their physician (disagree/uncertain = 0; agree = 1); belief that influenza vaccination protects against severe complications (disagree/uncertain = 0; agree = 1); concern that their children may contract influenza (not worried = 0; very worried = 1); concern that influenza may have serious consequences on their children’s health (not worried = 0; very worried = 1); belief that influenza vaccination is useful (not useful = 0; very useful = 1); having been informed about influenza vaccination by physicians (no = 0; yes = 1); need to receive additional information about influenza vaccination (no = 0; yes = 1). In Model 2, the independent variable at least one influenza vaccination in children in the four influenza seasons from 2018/2019 to 2021/2022 (no = 0; yes = 1) was also included. Results are presented as odds ratios (ORs) and 95% confidence intervals (CIs). All reported 𝑝 values are two-tailed and a value < 0.05 was considered statistically significant.

## 3. Results

### 3.1. Socio-Demographic and Anamnestic Characteristics of Parents and Children

In total, 927 questionnaires were sent out and 574 parents agreed to participate in the study with a response rate of 62%. The main characteristics of children and parents are described in Table 1. A large majority of responding parents were females (87.4%) and married/cohabitant (91.7%), and the average age was 38.5 years (range: 23–62). In less than half of the families (42.9%) there was at least one parent with a university degree, and in 33.3% at least one unemployed parent. At least one influenza vaccination uptake in the four influenza seasons from 2018/2019 to 2021/2022 was reported by 29.5% of parents and 18.1% documented a side effect after receiving the vaccination. An increase in the adherence to influenza immunization was found, ranging from 6% in the 2018/2019 season to 16.8% in the 2021/2022 season. Most of respondents (67.9%) had two or more children, with an average age of 4.6 years (range: <1–8), and 7.7% of children had at least one chronic condition. 

### 3.2. Parents’ Knowledge Regarding Influenza and Related Vaccination

Almost all parents were aware that influenza is transmitted by droplets (90.3%), but only 27.7% recognized that it is a vaccine-preventable disease, and 22.1% that influenza vaccination is recommended for 6 months to 6 years-old children. Moreover, only 31.4% of the respondents had an accurate knowledge about the main mandatory and recommended vaccinations for 6 months to 6 years-old children (Table 1).

### 3.3. Parents’ Attitudes Regarding Influenza and Related Vaccination

When parents were asked about their attitudes towards vaccination, 55% agreed or were uncertain that for their children it is better to acquire immunization through illness than through the vaccination, 51.9% believed that there are too many vaccinations for children, 74.4% agreed that influenza vaccination protects against severe complications, and 87.2% declared that they are confident on the information about vaccinations provided by their physician. Only 20.2% of parents were very concerned about their children’s risk of contracting influenza with an overall mean value of 5.3 out of a maximum score of 10, and 28.4% that influenza may have serious consequences on their children’s health. The mean total score indicating usefulness of vaccination to prevent influenza was 5.9, and 31.4% had a very favorable attitude (scores from 8 to 10). Moreover, 28.4% of parents were concerned about influenza vaccination side effects (Table 1).

### 3.4. Parents’ Behaviors about Vaccinations for Their Children 

The majority of parents (85.6%) reported that their child had had at least one visit by the primary care pediatrician/physician in the previous 12 months; specifically, 16.2% by the primary care physician and 78.2% by primary care pediatrician. Only 31.8% reported that their children had been vaccinated at least once in the four influenza seasons from 2018/2019 to 2021/2022; specifically, 52.4% of children with at least one chronic condition and 30.1% of healthy children. Moreover, an increase in adherence was found, ranging from 5.9% in the 2018/2019 to 18.9% in the 2021/2022 influenza season. The results of the multivariate logistic regression model showed that parents having received influenza vaccine at least once (OR = 7.29; 95% CI 3.79–14.05), having knowledge about the main mandatory and recommended vaccinations for 6 months to 6 years-old children (OR = 2.70; 95% CI 1.41–5.18), who knew that influenza can be prevented by vaccination (OR = 2.07; 95% CI 1.08–3.95), that influenza vaccination is recommended for healthy 6 months to 6 years-old children (OR = 3.42; 95% CI 1.67–6.99), who believed that influenza vaccination is useful (OR = 5.25; 95% CI 2.32–11.87), and have been informed about influenza vaccination by physicians (OR = 4.19; 95% CI 1.17–14.99) were significantly more likely to have vaccinated at least once their children in the 2018/2019–2021/2022 influenza seasons (Model 1 in Table 2). 

### 3.5. Parents’ Willingness to Vaccinate Their Children 

Almost half of parents (48.6%) declared their willingness to vaccinate their children in the upcoming 2022/2023 influenza season and, specifically, this willingness was expressed by 89.5% of parents who had ever vaccinated their children and by 29.2% who had never vaccinated their children in the four influenza seasons from 2018/2019 to 2021/2022, by 95% who had vaccinated their children in the previous 2021/2022 season, and by 37.8% who had not vaccinated their children in the previous 2021/2022 season. Among parents who expressed this intention, the most commonly reported reasons were being favorable to vaccinations (52.6%), considering influenza vaccination effective (47.4%) and safe (39.1%), having received a recommendation from the primary care pediatrician (40.9%), and considering the usefulness of the vaccination to distinguish influenza symptoms from those of COVID-19 (21.8%). Reasons for refusing to vaccinate their children in the upcoming 2022/2023 influenza season were lack of influenza vaccine recommendation from the primary care pediatrician (64.3%), considering their children not to be at risk (25.2%), and fear of adverse events (10.2%). The results of the multivariate logistic regression model showed that parents who had vaccinated at least once their children in the four influenza seasons from 2018/2019 to 2021/2022 (OR = 14.28; 95% CI 6.05–33.71), who did not know that influenza can be transmitted by droplets (OR = 0.23; 95% CI 0.07–0.77), believed that the influenza vaccination protects against severe complications (OR = 2.86; 95% CI 1.2–6.82), believed that influenza vaccine is useful (OR = 5.55; 95% CI 2.36–13.08), and believe that it is not better for children to acquire immunization through illness than through vaccination (OR = 6.23; 95% CI 3–12.96) had significantly higher odds to be willing to vaccinate their children in the upcoming 2022/2023 influenza season. Parents’ age demonstrated an impact on the intention to vaccinate their children in the upcoming 2022/2023 influenza season, since respondents aged 45 years or more (OR = 0.2; 95% CI 0.04–0.98) were less willing compared to those aged 35 years or less (Model 2 in Table 3).

### 3.6. Sources of Information

Overall, 96.6% of respondents had received information about influenza vaccination and 86.9% from physicians. Moreover, 47.7% declared to need additional information about influenza vaccination, and 91% expressed the preference to receive information from physicians.

## 4. Discussion

Although in Italy influenza vaccination in children with several underlying chronic medical conditions has long been recommended, only recently have healthy children 6 months to 6 years-old been included among the target groups for seasonal influenza vaccination. This is the first study exploring the extent of adherence to this new recommendation among parents of children of this age group, as well as the willingness to vaccinate their children in the following influenza season. The results of this investigation, which also explored determinants of prior influenza vaccination and intention to vaccinate children, may be useful for designing tailored strategies to enhance adherence to recommendations that are aimed at protecting not only children, but other subjects at risk of severe complications, such as adults with comorbidities and the elderly.

One of the main findings of this investigation is that knowledge that influenza vaccination is now recommended for healthy children has not yet spread among parents, since only slightly more than one fifth (22.1%) was aware about this opportunity for their children. This low awareness is coupled with incorrect beliefs related to influenza, with the large majority of respondents underestimating the risk for their children to contract influenza (79.2%) and to develop serious consequences (71.6%), as well as the usefulness of this vaccination, with only less than one third (31.4%) declaring that it is beneficial for children. 

Given these results on awareness and attitudes, it is not surprising that 31.8% of children had been vaccinated at least once in the preceding four influenza season (from 2018/2019 to 2021/2022), with an increasing trend from 2018/2019 to 2021/2022, with the highest adherence, involving 18.9% of children, found in 2021/2022. These findings suggest that most children had been vaccinated in the more recent seasons, probably as a consequence of the new recommendation on healthy children and of the co-occurrence of the COVID-19 pandemic, and it was confirmed by the age distribution of vaccinated children. Indeed, it was expected that older children (>3 years) would have been more likely to be vaccinated at least once in the preceding four seasons. Instead, while the frequency of influenza vaccination was 34.4% in younger children (≤3 years), it was 31.6% in those older (>3 years), suggesting that, regardless of age, adherence to influenza vaccination is a very recent phenomenon in healthy children. Data on influenza vaccination coverage in children have been explored in studies, which are heterogeneous regarding characteristics of the involved populations and methods used and timeframe, thus making difficult to compare data across studies. The findings of this study show a lower adherence compared to Flood et al. [22] who, relying on self-reporting of parents in a nationwide survey in USA, reported that 44.2% of children aged 2–12 years were vaccinated against influenza every year, and to Smith et al. [23] who documented parents’ self-reported uptake of the childhood influenza vaccine to be 52.8% in 2 to 7 years-old children in the 2015/2016 season in England. Similar trends were reported by D’Ambrosio et al. who, in a recent study in Italy, analyzed data from the Ministry of Health and found that in the period 2010/2011–2019/2020 the highest value of coverage was 4.5% in the 2 to 4 and 5 to 8 years-old children, whereas there was a substantial increase in the 2020/2021 season in all the age groups with 19% coverage in 2–4 years-old children, 13.1% in 5–8 years-old children, and 6.4% in <2 years-old children, this increase associated with a higher willingness from parents to vaccinate their children to protect them from influenza and facilitate differential diagnosis with COVID-19 [17]. Conversely, Fogel et al. in Germany revealed lower rates of influenza vaccination in 2020 (29.7%) compared with 2018 (34.2%) and 2019 (33.3%), suggesting that the COVID-19 pandemic might have influenced parental intent to obtain the influenza vaccine, or introduced barriers to healthcare access [26].

It is worthy of remark, however, that most of the children were visited by a primary care physician/pediatrician in the previous year, and the low adherence testifies that these professionals are not yet playing an effective role in promoting influenza vaccination to healthy children. Thus, further research is needed to investigate to what extent primary care physicians/pediatricians are aware and are promoting influenza vaccination in this target group. The influential role of physicians was also confirmed by the results of the model on determinants of adherence to influenza vaccination, which showed that information about vaccination by physicians predicted higher adherence, and by previous studies on influenza vaccination [22,27], as well as for other vaccinations in children [20,28,29,30].

As expected, greater knowledge and positive attitudes with respect to influenza vaccination, as well as past adherence of parents to this vaccination, predicted influenza uptake in children. Specifically, knowledge about vaccinations recommended to children, on the role of vaccination for the prevention of influenza, and on the new recommendation of influenza vaccination for children, were found to be associated with children’s uptake of influenza vaccination. Lack of knowledge was identified as a barrier to influenza vaccination in children in all risk groups and in the general public in a systematic review by Schmid et al. [31]. In the same review, numerous studies were appraised, in which past behavior was identified as a strong predictor of influenza vaccine acceptance in several population groups [31].

According to the results on willingness, there was propensity of parents to vaccinate their children at a higher rate in the 2022/2023 influenza season (48.6%), and this is encouraging, especially when coupled with the finding that 95% of those who had, and 37.8% of those who had not vaccinated their children in the previous 2021/2022 season, planned to adhere in the following year. However, a cautious approach should be maintained, since this result may not be related to a “real” increase, but rather reflects, at least partly, the gap that is commonly observed between intentions and health-related behaviors [32]. Nevertheless, the proportion of parents willing to vaccinate children in the upcoming 2022/2023 influenza season is within the large heterogeneity of the results reported in the literature, with the survey by Flood et al. that observed an intention to vaccinate ranging from 21.2% to 48.2% [22], slightly lower compared to the propensity to vaccinate for the next year (54.2%) reported by Goldman et al. in their survey on children’s presenting at the emergency care [33], and far lower than the 70% documented by Smith et al. for the 2016/2017 influenza season [23]. Moreover, hesitancy for influenza vaccines was reported to be 25.4% in parents participating in a nationwide survey in USA [34].

Confirmed by previous publications [22,23,33,35], key drivers of intention to adhere to the next influenza vaccination campaign for their children were recommendations by physicians/pediatricians, as well as past experience with influenza vaccination, and both factors may be included among the cues to action involved in parents’ decision-making on influenza vaccination for their children. Not surprisingly, other reasons or determinants of willingness pertained to perceived effectiveness and usefulness of the influenza vaccine, and to an overall positive attitude towards vaccinations, whereas barriers were represented by lack of perceived risk or threat of influenza for their children, and to perceived unsafety of the vaccine. Taken together, these factors may all be included in the framework of the Health Belief Model [36], which has frequently been involved in explaining health related intentions and behaviors. Many of these drivers and barriers have been reported in earlier literature on influenza vaccination in children [22,23,33,34,37].

It is of note that an increase in willingness to undergo influenza vaccination as a consequence of the COVID-19 pandemic was observed among the reasons to adhere, in order to distinguish influenza symptoms from those of COVID-19, and this motivation was also reported in the studies by D’Ambrosio et al. [17], and Goldman et al. [33], as well as in other population groups, including the general public [38] and healthcare workers [24,39,40].

Finally, the study results show that younger parents had higher propensity to vaccinate their children in the following year. To our knowledge, age of the parents has not been recognized in the literature as a determinant of intention to vaccinate with influenza vaccine, whereas age of children has sometimes been investigated, with younger children being more likely to be vaccinated [23]. Therefore, it may be tentatively hypothesized that our finding is related to the higher probability of younger parents to have, on average, younger children.

### Limitations

This study presents several limitations due to the study design. First, since this is a cross-sectional study, it is possible only to observe associations and not attribution of cause and effect due to temporal relationship between the determinants and the different outcomes of interest. Second, regarding attitudes, respondents may have been influenced to give the most “desirable” answers, and this may have produced overestimation of the positive attitudes toward the willingness to vaccinate their children in the upcoming 2022/2023 influenza season. Third, this study was conducted during the COVID-19 pandemic; therefore, at a time during which attention towards vaccines was high, and this could have had an influence on parents’ willingness and adherence to influenza vaccination. Forth, since previous children’s influenza vaccination uptake was investigated, a recall bias may have occurred, overestimating or underestimating this information. To contain these limitations the questionnaire was self-administered and, therefore, the confidentiality and anonymity of responses were guaranteed. Fifth, the majority of the sample was composed by mothers, and fathers’ opinions were not completely assessed. Making the decision to vaccinate children is a complex process, which should involve the participation of both parents. However, in our context, very often, the mother is the one who takes care of everything that concerns the health of children; therefore, we believe that the data on the willingness of influenza vaccination is reliable. Furthermore, some characteristics of the sample may give rise to issues of generalizability. Indeed, a large number of children (86%) had been visited by a physician in the previous 12 months, and 33% of respondents declared themselves to be unemployed. However, in Italy, every child is assigned a primary care pediatrician and it is common to make at least a control visit once a year. Moreover, since most of the respondents were mothers of young children, they might have been temporarily unemployed to take care of their children. Therefore, we believe that our results are generalizable at least to the southern Italian parent population. Finally, since the questionnaire was completed at home, parents could have retrieved information regarding the topic of the survey before responding to the questionnaire. Despite these limitations, a high response rate was achieved, and the findings of this study provide important insights and novel knowledge on a public health issue which has a relevant impact, not only on children’s health, but also on frail subjects of all ages. 

## 5. Conclusions

The present study documented that, although there has been an increase in children’s uptake of influenza vaccination in the 2018/2019 to 2021/2022 seasons, knowledge on new recommendations is not yet widespread, and the adherence to influenza vaccination among 6 months to 6 years-old children remains very low. However, an increase in the intention of parents to vaccinate their children in the upcoming 2022/2023 influenza season was revealed, particularly among those who had already been vaccinated. Since one of the most influential factors associated with the future receipt of influenza vaccination was found to be recommendations by physicians, efforts should be devoted to motivating healthcare professionals to promote influenza vaccination in children by educating parents on the risks correlated with the disease and the benefits of the vaccine for their children and for the entire community. 

## Figures and Tables

**Table 1 vaccines-10-01298-t001:** Main characteristics of parents participating in the survey, the associated influenza vaccination uptake, and willingness to vaccinate their children.

Characteristics	Total		Having Vaccinated at Least Once Their Children in the Four Influenza Seasons from 2018/2019 to 2021/2022		Willingness to Vaccinate Their Children in the Upcoming 2022/2023 Influenza Season	
Socio-Demographic and Anamnestic Characteristics	N	%	N	%	N	%
**Parents’ gender (573) ^a^**						
Male	72	12.6	20	31.8	26	39.4
Female	501	87.4	154	31.8	243	49.9
			X^2^ < 0.001, df = 1, *p* = 0.999		X^2^ = 2.57, df = 1, *p* = 0.109	
**Parents’ age, years (563) ^a^**	38.5 ± 6.25 (range: 23–62) ^b^					
≤35	182	32.3	46	26.9	86	48.9
36–40	169	30.1	56	34.2	86	52.4
41–45	146	25.9	48	33.6	67	47.5
>45	66	11.7	19	31.7	26	41.9
			X^2^ = 2.49, df = 3, *p* = 0.476		X^2^ = 2.13, df = 3, *p* = 0.546	
**Parents’ nationality (570) ^a^**						
Italian	539	94.6	167	32.4	252	48.4
Foreigners	31	5.4	6	20	15	51.7
			X^2^ = 2.00, df = 1, *p* = 0.157		X^2^ = 0.12, df = 1, *p* = 0.725	
**Marital status (570) ^a^**						
Married/cohabitant	523	91.7	161	31.9	247	48.8
Single parent	47	8.3	12	29.3	20	45.5
			X^2^ = 0.12, df = 1, *p* = 0.729		X^2^ = 0.18, df = 1, *p* = 0.669	
**Occupation (559) ^a^**						
Unemployed	186	33.3	47	26.6	82	45.8
Employed	373	66.7	125	34.8	184	50.8
			X^2^ = 3.72, df = 1, *p* = 0.054		X^2^ = 1.20, df = 1, *p* = 0.272	
**Education level (567) ^a^**						
University degree/master	243	42.9	96	40.5	122	51.5
Primary, middle and high school	324	57.1	76	24.9	143	46.1
			X^2^ = 14.96, df = 1, *p* < 0.001		X^2^ = 1.54, df = 1, *p* = 0.215	
**Parents having received influenza vaccine at least once in the four influenza seasons from 2018/2019 to 2021/2022 (566) ^a^**						
No	399	70.5	64	16.7	140	36.5
Yes	167	29.5	109	68.1	126	76.8
			X^2^ = 137.42, df = 1, *p* < 0.001		X^2^ = 74.98, df = 1, *p* < 0.001	
**Side effects after influenza vaccination (165) ^ao^**						
No	135	81.8	89	69.9	105	79.6
Yes	30	18.2	18	62.1	19	63.3
			X^2^ = 0.52, df = 1, *p* = 0.471		X^2^ = 3.58, df = 1, *p* = 0.059	
**Number of children, (ordinal) (569) ^a^**						
1	183	32.1	45	25.6	90	50.3
2	294	51.7	98	34.8	138	48.8
≥3	92	16.2	29	33.3	38	44.2
			X^2^ = 4.38, df = 2, *p* = 0.112		X^2^ = 0.88, df = 2, *p* = 0.646	
**Children’s age, years (552) ^a^**	4.6 ± 1.33 (range: 0–8)^b^					
≤3	102	18.5	33	34.4	52	53.1
>3	450	81.5	137	31.6	206	47.4
			X^2^ = 0.27, df = 1, *p* = 0.604		X^2^ = 1.04, df = 1, *p* = 0.307	
**Children with at least one chronic disease (571) ^a^**						
No	527	92.3	152	30.1	243	47.7
Yes	44	7.7	22	52.4	25	59.5
			X^2^ = 8.88, df = 1, *p* = 0.003		X^2^ = 2.16, df = 1, *p* = 0.142	
**Children having had at least one visit by the primary care pediatrician/physician in the previous 12 months (568) ^a^**						
No	82	14.4	26	33.8	40	51.9
Yes	486	85.6	146	31.3	227	48.2
			X^2^ = 0.19, df = 1, *p* = 0.662		X^2^ = 0.37, df = 1, *p* = 0.541	
**Knowledge regarding influenza and related vaccination**	**N**	**%**	**N**	**%**	**N**	**%**
Knowledge of influenza modes of transmission (564) ^a^						
Droplet	509	90.3	157	31.9	239	48.6
Other	55	9.7	16	32	28	53.9
			X^2^ < 0.001, df = 1, *p* = 0.997		X^2^ = 0.52, df = 1, *p* = 0.470	
**Knowledge that influenza can be prevented by vaccination (563) ^a^**						
No	407	72.3	103	26.6	163	41.9
Yes	156	27.7	71	46.4	105	67.7
			X^2^ = 19.83, df = 1, *p* < 0.001		X^2^ = 29.61, df = 1, *p* < 0.001	
**Knowledge that influenza vaccination is recommended** **for healthy 6 months to 6 years-old children (539) ^a^**						
No	420	77.9	107	26.4	186	45.3
Yes	119	22.1	62	52.3	77	66.4
			X^2^ = 29.46, df = 1, *p* < 0.001		X^2^ = 16.15, df = 1, *p* < 0.001	
**Accurate knowledge about the main mandatory and recommended vaccinations for** **6 months to 6 years-old children (401) ^a^**						
No	275	68.6	62	23.5	113	41.7
Yes	126	31.4	67	54.9	82	66.1
			X^2^ = 37.05, df = 1, *p* < 0.001		X^2^ = 20.32, df = 1, *p* < 0.001	
**Attitudes towards influenza and related vaccination**	**N**	**%**	**N**	**%**	**N**	**%**
**Belief that it is better for children to acquire immunization through illness than through vaccination (553) ^a^**						
Disagree	249	45	97	40.3	166	68
Agree/Uncertain	304	55	72	24.6	95	32.2
			X^2^ = 15.02, df = 1, *p* < 0.001		X^2^ = 68.64, df = 1, *p* < 0.001	
**Belief that there are too many vaccinations for children (549) ^a^**						
Disagree	264	48.1	101	39.2	144	55
Agree/Uncertain	285	51.9	67	24.7	116	42.3
			X^2^ = 12.69, df = 1, *p* < 0.001		X^2^ = 8.55, df = 1, *p* = 0.003	
**Confidence in the information** **about vaccinations provided by their physician (561) ^a^**						
Disagree/Uncertain	72	12.8	12	17.6	18	25.7
Agree	489	87.2	159	33.7	246	51.9
			X^2^ = 7.07, df = 1, *p* = 0.008		X^2^ = 16.74, df = 1, *p* < 0.001	
**Belief that the influenza vaccine protects against severe complications (558) ^a^**						
Disagree/Uncertein	143	25.6	28	20.1	38	27.1
Agree	415	74.4	142	35.5	225	55.8
			X^2^ = 11.27, df = 1, *p* = 0.001		X^2^ = 34.24, df = 1, *p* < 0.001	
**Concerns about influenza and related vaccination**	**N.**	**%**	**N.**	**%**	**N.**	**%**
**Concern that their children may contract influenza (565) ^a^**	5.3 ± 2.54 (range: 1–10) ^b^					
Not worried	451	79.8	130	29.8	197	44.5
Very worried	114	20.2	44	40.7	72	67.9
			X^2^ = 4.81, df = 1, *p* = 0.028		X^2^ = 18.83, df = 1, *p* < 0.001	
**Concern that influenza may have serious consequences on their children’s health (563) ^a^**	5.8 ± 2.79 (range: 1–10) ^b^					
Not worried	403	71.6	120	30.5	180	45.3
Very worried	160	28.4	53	35.1	89	58.6
			X^2^ = 1.05, df = 1, *p* = 0.306		X^2^ = 7.68, df = 1, *p* = 0.006	
**Belief that influenza vaccination is useful (560) ^a^**	5.9 ± 2.76 (range: 1–10) ^b^					
Not useful	384	68.6	70	18.7	117	31.4
Very useful	176	31.4	103	62.1	151	86.8
			X^2^ = 99.55, df = 1, *p* < 0.001		X^2^ = 145.8, df = 1, *p* < 0.001	
**Sources of information**	**N**	**%**	**N**	**%**	**N**	**%**
**Having been informed about influenza vaccination by physicians (556) ^a^**						
No	73	13.1	18	25.4	33	45.2
Yes	483	86.9	154	32.8	232	49.3
			X^2^ = 1.56, df = 1, *p* = 0.211		X^2^ = 0.42, df = 1, *p* = 0.519	
**Need to receive additional information about influenza vaccination (560) ^a^**						
No	293	52.3	92	32.1	136	47.4
Yes	267	47.7	81	31.9	132	50.8
			X^2^ = 0.002, df = 1, *p* = 0.967		X^2^ = 0.62, df = 1, *p* = 0.429	

^a^ Number of each item may not add up to total number of study population due to missing values. In parenthesis the number of respondents to each item. ^b^ Mean ± Standard deviation (range). ^o^ Only among those who had received influenza vaccine.

**Table 2 vaccines-10-01298-t002:** Multivariate logistic regression analyses to characterize factors associated with having vaccinated at least once their children in the four influenza seasons from 2018/2019 to 2021/2022.

° Model 1. Having Vaccinated at Least Once Their Children in the Four Influenza Seasons from 2018/2019 to 2021/2022				
Log likelihood = −133.69, χ^2^ = 155.72 (14 df), *p* < 0.0001, No. of Observations = 332				
Variable	OR		95% CI	*p*
Parents having received influenza vaccine at least once in the four influenza seasons from 2018/2019 to 2021/2022				
No	1.00 *			
Yes	7.29		3.79–14.05	<0.001
Knowledge about the main mandatory and recommended vaccinations for 6 months to 6 years-old children				
No	1.00 *			
Yes	2.70		1.41–5.18	0.003
Knowledge that influenza can be prevented by vaccination				
No	1.00 *			
Yes	2.07		1.08–3.95	0.028
Knowledge that influenza vaccination is recommended for a healthy 6 months to 6 years-old children				
No	1.00 *			
Yes	3.42		1.67–6.99	<0.001
Belief that influenza vaccination is useful				
Not useful	1.00 *			
Very useful	5.25		2.32–11.87	<0.001
Had been informed about influenza vaccination by physicians				
No	1.00 *			
Yes	4.19		1.17–14.99	0.028
Parents’ age, years				
≤35	1.00 *			
36–40	0.74		0.38–1.44	0.373
41–45	Backward elimination			
>45	Backward elimination			
Number of children, ordinal (1 = 1; 2 = 3; ≥3 = 3)	1.31		0.83–2.09	0.251
Children with at least one chronic disease				
No	1.00 *			
Yes	1.82		0.55–6.01	0.326
Belief that it is better for children to acquire immunization through illness than through vaccination				
Agree/Uncertain	1.00 *			
Disagree	0.61		0.29–1.26	0.180
Belief that there are too many vaccinations for children				
Agree/Uncertain	1.00 *			
Disagree	1.47		0.77–2.82	0.247
Confidence in the information about vaccinations provided by their physician				
Disagree/Uncertain	1.00 *			
Agree	1.68		0.54–5.25	0.370
Belief that the influenza vaccine protects against severe complications				
Disagree/Uncertain	1.00 *			
Agree	1.63		0.66–4.03	0.293
Concern that influenza may have serious consequences on their children’s health				
Not worried	1.00 *			
Very worried	0.48		0.21–1.06	0.070

* Reference category. ° The following variables were deleted by the backward elimination procedure: gender, education level, marital status, age group 41–45 years, age group > 45 years, occupation, children’s age, knowledge of influenza modes of transmission, concern that their children may contract influenza, children having had at least one visit by the primary care pediatrician/physician in the previous 12 months, need to receive additional information about influenza vaccination.

**Table 3 vaccines-10-01298-t003:** Multivariate logistic regression analyses to characterize factors associated with willingness to vaccinate their children in the upcoming 2022/2023 influenza season.

° Model 2. Willingness to Vaccinate Their Children in the Upcoming 2022/2023 Influenza Season				
Log Likelihood = −127.48, χ^2^ = 203.75( 12 df), *p* < 0.0001, No. of Observations = 331				
Variable	OR		95% CI	*p*
Parents’ age, years				
≤35	1.00 *			
36–40	Backward elimination			
41–45	Backward elimination			
>45	0.2		0.04–0.98	0.048
Knowledge of influenza modes of transmission				
Other	1.00 *			
Droplet	0.23		0.07–0.77	0.017
It is better for children to acquire immunization through illness than through vaccination				
Agree/Uncertain	1.00 *			
Disagree	6.23		3–12.96	<0.001
Belief that influenza vaccination protects against severe complications				
Disagree/Uncertain	1.00 *			
Agree	2.86		1.2–6.82	0.018
Belief that influenza vaccination is useful				
Not useful	1.00 *			
Very useful	5.55		2.36–13.08	<0.001
Had vaccinated at least once their children in the four influenza seasons from 2018/2019 to 2021/2022				
No	1.00 *			
Yes	14.28		6.05–33.71	<0.001
Parents having received influenza vaccine at least once in the four influenza seasons from 2018/2019 to 2021/2022				
No	1.00 *			
Yes	2.09		0.96–4.56	0.064
Number of children (1 = 1; 2 = 3; ≥3 = 3)	0.75		0.46–1.21	0.241
Children with at least one chronic disease				
No	1.00 *			
Yes	0.35		0.09–1.48	0.155
Belief that there are too many vaccinations for children				
Agree/Uncertain	1.00 *			
Disagree	0.54		0.27–1.05	0.070
Concern that their children may contract influenza				
Not worried	1.00 *			
Very worried	1.88		0.81–4.37	0.141
Having been informed about influenza vaccination by physicians				
No	1.00 *			
Yes	0.62		0.2–1.86	0.389

* Reference category. ° The following variables were deleted by the backward elimination procedure: gender, education level, marital status, age group 41–45 years, age group > 45 years, occupation, children’s age, children having had at least one visit by the primary care pediatrician/physician in the previous 12 months, knowledge that influenza can be prevented by vaccination, knowledge that influenza vaccination is recommended for healthy 6 months to 6 years-old children, accurate knowledge about the main mandatory and recommended vaccinations for 6 months to 6 years-old children, confidence in the information about vaccinations provided by their physician, concern that influenza may have serious consequences on their children’s health, need to receive additional information about influenza vaccination.

## Data Availability

The data presented in this study are available on request from the corresponding author.

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
