# Peer review of "Surveying Parents’ Awareness and Adherence to Influenza Vaccination Recommendations in Children in Southern Italy"

_vaccines, 2022, doi:10.3390/vaccines10081298_

Round 1
Reviewer 1 Report
This is an important and excellent article and clearly presented. Well done!
39. Moreover, there is consistent evidence that children play a leading role in propagating influenza A epidemics to other age-groups of the population [5,6], including older 40 people, who are at higher risk of severe influenza complications [7].
[this is the key statement motivating your study. Please provide data from key studies]
86 A self-administered structured questionnaire was outlined and developed adopting the experience of previously published studies.
[do you have any reliability data from previous uses of your form? This will be helpful to future researchers]
The questions were grouped into four sections: 1) socio-demographic (gender, age, 88 marital status, level of education, working activity, nationality, number of children, age 89 and gender of the child) and anamnestic characteristics of parents and children (influ- 90 enza vaccination uptake of parent in the 2018/19-2021/22 seasons and eventual 91 side-effects, chronic conditions of the child, and visits to physician/pediatrician in the 92 previous year); 2) knowledge on recommended childhood vaccinations, including in- 93 fluenza, and on modes of transmission and prevention of influenza, as well as on popu- 94 lation groups who are recommended to receive influenza vaccination; 3) attitudes and 95 behaviours related to childhood vaccinations overall (perceptions of benefits and risks of 96 Vaccines 2022, 10, x FOR PEER REVIEW 3 of 16 vaccinations compared to the related diseases and adherence to childhood vaccinations’ 97 recommendations for their children), and specifically to influenza (perception of risks 98 related to the disease and usefulness of the vaccines, having vaccinated their children in 99 the 2018/19-2021/22 seasons, intention to vaccinate them in the following season, and 100 reasons related to their intention); 4) sources of information on influenza vaccination and 101 need of additional information.
[This is complicated data, especially parts 2 and 3. You have described lower down in the manuscript how you summarised the answers but it is key to see how the questions are structured so they are easily answerable by parents. Can you please provide your questionnaire as Supplemental Table for the web for future researchers?]
121 A pilot study among 50 parents was carried out to evaluate the readability, clarity, and correct flow of the items.
[briefly what were the results – what changes were need if any? of interest to future researchers]
Limitations:
[Thank you for the comprehensive limitations section. Please add 86% visit with primary care physician, 33% unemployed, 57% university degree so unusual sample. Pleas add to whom can your results be generalised].
Follow up
[Will you now present your results and advice to the Campania authorities?]
English text is excellent.
Author Response
- Moreover, there is consistent evidence that children play a leading role in propagating influenza A epidemics to other age-groups of the population [5,6], including older 40 people, who are at higher risk of severe influenza complications [7].
[this is the key statement motivating your study. Please provide data from key studies]
As suggested, data on the role of children have been added.
86 A self-administered structured questionnaire was outlined and developed adopting the experience of previously published studies.
[do you have any reliability data from previous uses of your form? This will be helpful to future researchers]
As suggested, we have referenced publications where parts of the questionnaire have been used.
The questions were grouped into four sections: 1) socio-demographic (gender, age, marital status, level of education, working activity, nationality, number of children, age and gender of the child) and anamnestic characteristics of parents and children (influenza vaccination uptake of parent in the 2018/19-2021/22 seasons and eventual side-effects, chronic conditions of the child, and visits to physician/pediatrician in the previous year); 2) knowledge on recommended childhood vaccinations, including influenza, and on modes of transmission and prevention of influenza, as well as on population groups who are recommended to receive influenza vaccination; 3) attitudes and behaviours related to childhood vaccinations overall (perceptions of benefits and risks of vaccinations compared to the related diseases and adherence to childhood vaccinations’ recommendations for their children), and specifically to influenza (perception of risks related to the disease and usefulness of the vaccines, having vaccinated their children in the 2018/19-2021/22 seasons, intention to vaccinate them in the following season, and 100 reasons related to their intention); 4) sources of information on influenza vaccination and 101 need of additional information.
[This is complicated data, especially parts 2 and 3. You have described lower down in the manuscript how you summarised the answers but it is key to see how the questions are structured so they are easily answerable by parents. Can you please provide your questionnaire as Supplemental Table for the web for future researchers?]
As suggested, we have simplified the description of the questionnaire, and have included it as a Supplementary material.
121 A pilot study among 50 parents was carried out to evaluate the readability, clarity, and correct flow of the items.
[briefly what were the results – what changes were need if any? of interest to future researchers]
As suggested, we have included the only modification that was necessary after the completion of the pilot study.
Limitations:
[Thank you for the comprehensive limitations section. Please add 86% visit with primary care physician, 33% unemployed, 57% university degree so unusual sample. Pleas add to whom can your results be generalised].
In response to this point there was a typo in Table 1 and the 57% refers to the 324 subjects who had lower than university degree, whereas 42.9% refers to the 243 subjects with university. We have corrected this in the Table and in the text, but the sample remains unusual, and, as suggested, we have discussed the generalizability of the results in the limitations of the study.
Follow up
[Will you now present your results and advice to the Campania authorities?]
We hope we will be able to send them the published manuscript, as it would give strength to our data and be of more interest to the stakeholders!
Reviewer 2 Report
Thank you that you give me the opportunity to review this manuscript “
Surveying parents’ awareness and adherence to influenza vaccination recommendations in healthy children”.
The authors presented an interesting theme regards the influenza vaccination among children (6 mo-6 yr) in Campania, the group where this vaccination is still infrequent.
Some of the comments on this manuscript.
Title In the title it should be added that this concerns of southern Italy (or in Campania, in Italy) . The authors studied one small region of southern Italy, a small population.
Line 181- Authors write ….Moreover, only 31.4% of the respondents had an accurate knowledge about the main mandatory and recommended vaccinations for 6 months - 6 years old children. but the title mentioned only flu vaccination, it is worth adding others
Line 187- in Results authors write …51.9% believed that there are too many vaccinations for children…. but the title mentioned only flu vaccination, it is worth adding others
In tabl 2 Knowledge about the main mandatory and recommended vaccinations for 6 months - 6 years old children
In title is …. in healthy children, it is not true, because in table 1 Children with at least one chronic disease answer is 44.
I understand why the authors used "in healthy children", because it is explained in lines 47-49.
Unfortunately, the appearance of this in the title sounds a bit strange, as if it is possible to vaccinate sick children. This can mislead potential readers.
1.Introduction
Line 45: In Italy every year the Ministry of Health… . Who pays for the influenza vaccine? a patient or the Ministry of Health. Flu vaccines payment may have changed in subsequent seasons.
Line 54: What is the flu vaccination coverage in Italy? Before the pandemic COVID-19 and nowadays (2020/2021 season). And especially during the seasons in which the study was conducted.
In the introduction, the authors omitted an important point.
The authors did not describe the coverage of the influenza vaccine in the general population in the region Campania in which they conducted the study. And what is the percentage compared to the population of the whole of Italy?
Because if Campania is the region with a high flu vaccination rate, parents will also willingness to vaccine their children, otherwise, the rate will be low.
Line 202: an increase in the adherence was found, ranging from 5.9% in the 2018/19 to 18.9% in the 2021/22 influenza season
In addition, free (or partially paid) flu vaccines have been introduced in many countries during the COVID-19 pandemic. What does it look like in Italy ???
The COVID-19 pandemic increased interest in vaccinations not only among the adult population but also among parents who were previously reluctant to vaccinate. What does it look like in Italy ???
In keywords is “ vaccination coverage” but there is nothing about it in the introduction.
In addition, the introduction lacks information about the flu vaccine for children, which is completely different from that for adults, and when it was introduced for use in children in Italy. This is important because the short period of its availability means that parents simply cannot know about it yet. Please add it.
2.Material and Methods
2.1.
Line 63- This survey study was undertaken from March 15 to June 30 2022, and involved 63 parents of 6 months-6 years old children. How could a child who e.g.is 6 months old in June 2022 be vaccinated against the flu? Influenza vaccinations in Europe usually come in the fall.
Therefore, the division into groups of children up to 3 years old and over 3 years old is strange. A nearly three-year-old child had the opportunity to be vaccinated twice, and a six-month-old child never.
Line 66- Schools and kindergartens were randomly selected from the list of all public and private schools in the Naples metropolitan area. So please specify exactly- how many there are "all" schools and kindergartens
Line 68- What does it mean? ……and all eligible parents of children attending the selected structures were potentially included in the study.
what were the inclusion and exclusion criteria for the parents' study?
Line 69- The sample size was calculated by using single population proportion formula with the assumption that 50% of parents had vaccinated their children against influenza, using a confidence level of 95%, a margin of error at 5%, and considering a response rate of 60%..
Please provide the References on which the authors made this assumption (that 50% of parents had vaccinated their children against influenza), and provide the formula for how it was calculated. The description should be included in the supplementary.
Line 79- …..that confidentiality of responses would be granted, and were invited to fill and return the questionnaire and the signed consent form. To improve the response rate several follow-up reminders were provided by the schools’ directors to the selected parents.
If you want to keep the survey confidential, you cannot ask for signing the survey.
It is not surprising then that the directors had to remind the parents several times and there are many answers in missed surveys.
All the more so because in March 2019 in Italy, the law on the form of compulsory certain vaccinations was passed, which means that children whose parents present a vaccination card are admitted to kindergartens. The rules on vaccinating older children have also been tightened. and parents of unvaccinated children aged 6-16 may be fined up to EUR 500. Hence the afraid of parents to complete all answers the survey.
Line 79- …..that confidentiality of responses would be granted, and were invited to fill and return the questionnaire and the signed consent form. But in Line 366 is- To contain these limitations the questionnaire was self-administered and, therefore, the confidentiality and anonymity of responses were guaranteed.
In the manuscript are discrepancies. So, it was anonymous or not??
2.2 Survey instrument
Line 86- A self-administered structured questionnaire was outlined and developed adopting the experience of previously published studies. Please add references.
Line 115- Childhood vaccinations ’uptake was measured by marking a list of several vaccinations, and for influenza by marking the list of the 2018 / 19-2021 / 22 seasons.
The seasons 2018/19 - 2021/22, means 2 vaccination seasons before the COVID-19 pandemic, and during the COVID-19 pandemic. or Are there four seasons from 2018/19, 2019/2020, 2020/21 and 2021/22?
Or like the authors write in the end of the manuscript from … to
3. Results.
Table 1 What is the difference between cohabitation and unmarried. Cohabitation is also unmarried, and divorced also can live with a partner (and it is a cohabitant). Please explain it.
And again, why ask marial status in a survey if you are not using the results in multivariate logistic regression analyses- table 2 ? These are very personal questions, especially if you must sign a survey.
Line 160- …and 574 parents agreed to participate in…
In table
Male |
72 |
|
|
|
|
|
|
Female 501 it is total 573 (where is one parent ?) age parents 182+169+146+66=563, and where are the rest of the parents? and so on,…. each of the survey points in table 1 does not have a correct number. even Children's age, 102 + 450 = 552, 22 children are missing ( 22 parents).
|
|
||||||
In Table 1 with more than 20 questions from the survey, none of the items has the correct number of completed surveys.
Such surveys should not be included in the statistical calculations. Lots of gaps in answers affect the results of multi-regression, especially if the population is small.
Hence, if the survey is anonymous, no signature is taken. Women e.g. usually avoid giving age, and men e.g. hide lack of knowledge .
-Tabl 2. No. of obs = 332 , what does mean obs??
-Table 2 should be divided into two tables, because it is now very big.
Why did not the parents' education and occupation were taken into account in the multivariate logistic regression calculations in table 2? These are factors that influence the decision to vaccinate.
Line 269- that only 31.8% of children had been vaccinated at least once in the preceding four influenza season (2018/19-2021/22), it is very good result, especially that some of children were too young in June 2022 be vaccinated against the flu.
Authors cite their publications too often in the references.
Author Response
Title In the title it should be added that this concerns of southern Italy (or in Campania, in Italy). The authors studied one small region of southern Italy, a small population.
As suggested, we have specified in the title that the survey refers to southern Italy
Line 181- Authors write ….Moreover, only 31.4% of the respondents had an accurate knowledge about the main mandatory and recommended vaccinations for 6 months - 6 years old children. but the title mentioned only flu vaccination, it is worth adding others
Line 187- in Results authors write …51.9% believed that there are too many vaccinations for children…. but the title mentioned only flu vaccination, it is worth adding others
In tabl 2 Knowledge about the main mandatory and recommended vaccinations for 6 months - 6 years old children
In response to this point, our main focus was influenza vaccination in children; however, to have insight in the interest of the investigated population on this topic overall knowledge and attitudes on vaccinations was considered useful to test their interest also on influenza vaccination. As of your suggestion we have now included this secondary objective in the Introduction.
In title is …. in healthy children, it is not true, because in table 1 Children with at least one chronic disease answer is 44.
I understand why the authors used "in healthy children", because it is explained in lines 47-49.
Unfortunately, the appearance of this in the title sounds a bit strange, as if it is possible to vaccinate sick children. This can mislead potential readers.
As suggested, to avoid misunderstanding, we have eliminated the word “healthy” from the title.
1.Introduction
Line 45: In Italy every year the Ministry of Health… . Who pays for the influenza vaccine? a patient or the Ministry of Health. Flu vaccines payment may have changed in subsequent seasons.
As suggested, we have included policies related to influenza vaccination in Italy and how they evolved in the last years.
Line 54: What is the flu vaccination coverage in Italy? Before the pandemic COVID-19 and nowadays (2020/2021 season). And especially during the seasons in which the study was conducted.
As suggested, data on flu vaccination coverage in Italy have been included.
In the introduction, the authors omitted an important point.
The authors did not describe the coverage of the influenza vaccine in the general population in the region Campania in which they conducted the study. And what is the percentage compared to the population of the whole of Italy?
Because if Campania is the region with a high flu vaccination rate, parents will also willingness to vaccine their children, otherwise, the rate will be low.
As suggested, data on vaccination coverage in Campania region and comparisons with the whole of Italy have been added.
Line 202: an increase in the adherence was found, ranging from 5.9% in the 2018/19 to 18.9% in the 2021/22 influenza season
In addition, free (or partially paid) flu vaccines have been introduced in many countries during the COVID-19 pandemic. What does it look like in Italy ???
As suggested, we have described flu vaccination policies in the context of COVID-19 pandemic
The COVID-19 pandemic increased interest in vaccinations not only among the adult population but also among parents who were previously reluctant to vaccinate. What does it look like in Italy ???
As suggested, we have described vaccination attitudes of parents in the context of COVID-19 pandemic.
In keywords is “ vaccination coverage” but there is nothing about it in the introduction.
As suggested, we have eliminated “vaccination coverage” from the keywords.
In addition, the introduction lacks information about the flu vaccine for children, which is completely different from that for adults, and when it was introduced for use in children in Italy. This is important because the short period of its availability means that parents simply cannot know about it yet. Please add it.
As suggested, we have described flu vaccines available for children.
2.Material and Methods
2.1.
Line 63- This survey study was undertaken from March 15 to June 30 2022, and involved 63 parents of 6 months-6 years old children. How could a child who e.g.is 6 months old in June 2022 be vaccinated against the flu? Influenza vaccinations in Europe usually come in the fall.
In response to this point, children who could not be vaccinated in the last season was not included in the study; therefore included children could have at least received one influenza vaccination. To avoid misunderstanding, we have clarified this point in the methods section.
Therefore, the division into groups of children up to 3 years old and over 3 years old is strange. A nearly three-year-old child had the opportunity to be vaccinated twice, and a six-month-old child never.
In response to this point, we agree that the older the child the higher the opportunity to have been vaccinated at least once, whereas all included children had at least one opportunity to be vaccinated. However, in our results 34.4% of children < 3 years and 31.6% of those > 3 years had been vaccinated at least once; therefore, as of your suggestion, we have now discussed this point, arguing that probably our results are mostly influenced by the higher adherence in the most recent seasons, regardless of age.
Line 66- Schools and kindergartens were randomly selected from the list of all public and private schools in the Naples metropolitan area. So please specify exactly- how many there are "all" schools and kindergartens
As suggested, we have included the total number of nurseries and kindergartens.
Line 68- What does it mean? ……and all eligible parents of children attending the selected structures were potentially included in the study.
what were the inclusion and exclusion criteria for the parents' study?
In response to this point, we meant that all parents who met the inclusion criteria were invited to participate, without random sampling. To avoid misunderstanding, we have rephrased the sentence, and specified the inclusion criteria.
Line 69- The sample size was calculated by using single population proportion formula with the assumption that 50% of parents had vaccinated their children against influenza, using a confidence level of 95%, a margin of error at 5%, and considering a response rate of 60%..
Please provide the References on which the authors made this assumption (that 50% of parents had vaccinated their children against influenza), and provide the formula for how it was calculated. The description should be included in the supplementary.
As suggested references have been provided, and formula and calculation of sample size have been added in a supplementary file.
Line 79- …..that confidentiality of responses would be granted, and were invited to fill and return the questionnaire and the signed consent form. To improve the response rate several follow-up reminders were provided by the schools’ directors to the selected parents.
If you want to keep the survey confidential, you cannot ask for signing the survey.
It is not surprising then that the directors had to remind the parents several times and there are many answers in missed surveys.
In response to this point, parents received the questionnaire and the consent form as two separated documents. They signed the consent form, which was separated from the questionnaire, which was not signed. To avoid misunderstanding we have made this point clearer in the text.
All the more so because in March 2019 in Italy, the law on the form of compulsory certain vaccinations was passed, which means that children whose parents present a vaccination card are admitted to kindergartens. The rules on vaccinating older children have also been tightened. and parents of unvaccinated children aged 6-16 may be fined up to EUR 500. Hence the afraid of parents to complete all answers the survey.
See previous response.
Line 79- …..that confidentiality of responses would be granted, and were invited to fill and return the questionnaire and the signed consent form. But in Line 366 is- To contain these limitations the questionnaire was self-administered and, therefore, the confidentiality and anonymity of responses were guaranteed.
In the manuscript are discrepancies. So, it was anonymous or not??
See previous response
2.2 Survey instrument
Line 86- A self-administered structured questionnaire was outlined and developed adopting the experience of previously published studies. Please add references.
As suggested, we have added references of previous studies.
Line 115- Childhood vaccinations ’uptake was measured by marking a list of several vaccinations, and for influenza by marking the list of the 2018 / 19-2021 / 22 seasons.
The seasons 2018/19 - 2021/22, means 2 vaccination seasons before the COVID-19 pandemic, and during the COVID-19 pandemic. or Are there four seasons from 2018/19, 2019/2020, 2020/21 and 2021/22?
Or like the authors write in the end of the manuscript from … to
In response to this point we refer to the four influenza seasons from 2018/19 to 2021/22. To avoid misunderstanding we have always repeated that they are four seasons from… to….in the text.
- Results.
Table 1 What is the difference between cohabitation and unmarried. Cohabitation is also unmarried, and divorced also can live with a partner (and it is a cohabitant). Please explain it.
And again, why ask marial status in a survey if you are not using the results in multivariate logistic regression analyses- table 2 ? These are very personal questions, especially if you must sign a survey.
In response to this point, marital status was used as a proxy to distinguish single parent from two parents’ families, that is why married and cohabitant were collapsed. We have now accordingly modified the text.
Line 160- …and 574 parents agreed to participate in…
In table
Male 72
Female 501
it is total 573 (where is one parent ?)
age parents 182+169+146+66=563, and where are the rest of the parents?
and so on,….
each of the survey points in table 1 does not have a correct number.
even Children's age, 102 + 450 = 552,
22 children are missing ( 22 parents).
In Table 1 with more than 20 questions from the survey, none of the items has the correct number of completed surveys.
Such surveys should not be included in the statistical calculations. Lots of gaps in answers affect the results of multi-regression, especially if the population is small.
Hence, if the survey is anonymous, no signature is taken. Women e.g. usually avoid giving age, and men e.g. hide lack of knowledge .
In response to this point, we have specified in the table that, as expected in self-administered surveys, in the different items there could be some missing responses, giving a different total for each item. To avoid misunderstanding, we have now put in parenthesis the number of respondents to each item.
-Tabl 2. No. of obs = 332 , what does mean obs??
In response to this point, Obs is for observations, and we have now written the entire word
-Table 2 should be divided into two tables, because it is now very big.
As suggested, Table 2 has been split in Table 2 and Table 3.
Why did not the parents' education and occupation were taken into account in the multivariate logistic regression calculations in table 2? These are factors that influence the decision to vaccinate.
In response to this point, in the Statistical analysis section the variables that were included in the Models have been described and parents’ education and occupation were among them. However, they were eliminated from the stepwise procedure by backward elimination. To avoid misunderstanding, we have now specified it in the legend of the Tables reporting the results of the Models (Table 2 and Table 3).
Line 269- that only 31.8% of children had been vaccinated at least once in the preceding four influenza season (2018/19-2021/22), it is very good result, especially that some of children were too young in June 2022 be vaccinated against the flu.
As suggested, we have discussed this point in the Discussion section.
Authors cite their publications too often in the references.
As suggested, we have reduced the number of self-citations.
Round 2
Reviewer 2 Report
The authors have satisfactorily answered the questions asked and revised the manuscript.